# Performance Enhancement of Electrospun IGZO-Nanofiber-Based Field-Effect Transistors with High-*k* Gate Dielectrics through Microwave Annealing and Postcalcination Oxygen Plasma Treatment

**DOI:** 10.3390/nano10091804

**Published:** 2020-09-10

**Authors:** Seong-Kun Cho, Won-Ju Cho

**Affiliations:** Department of Electronic Materials Engineering, Kwangwoon University, 20, Gwangun-ro, Nowon-gu, Seoul 01897, Korea; whtjdrms98@gmail.com

**Keywords:** indium gallium zinc oxide, nanofiber, field-effect transistor, microwave annealing, oxygen plasma treatment

## Abstract

We investigated the effects of various high-*k* gate dielectrics as well as microwave annealing (MWA) calcination and a postcalcination oxygen plasma treatment on the electrical properties and stability of electrospun indium gallium zinc oxide (IGZO)-nanofiber (NF)-based field-effect transistors (FETs). We found that the higher the dielectric constant of the gate dielectric, the better the electric field is transferred, resulting in the better performance of the IGZO NF FET. In addition, the MWA-calcined IGZO NF FET was superior to the conventional furnace annealing-calcined device in terms of the electrical properties of the device and the operation of resistor-loaded inverter, and it was proved that the oxygen plasma treatment further improved the performance. The results of the gate bias temperature stress test confirmed that the MWA calcination process and postcalcination oxygen plasma treatment greatly improved the stability of the IGZO NF FET by reducing the number of defects and charge traps. This verified that the MWA calcination process and oxygen plasma treatment effectively remove the organic solvent and impurities that act as charge traps in the chemical analysis of NF using X-ray photoelectron spectroscopy. Furthermore, it was demonstrated through scanning electron microscopy and ultraviolet-visible spectrophotometer that the MWA calcination process and postcalcination oxygen plasma treatment also improve the morphological and optical properties of IGZO NF.

## 1. Introduction

The technologies for manufacturing modern electronic devices that exhibit high stretchability, flexibility, and transparency are attracting much attention owing to the increased demand for such devices in recent years [1]. In this regard, high-performance multifunctional devices based on nanomaterials such as nanofibers (NFs), nanowires, nanoparticles, and nanoribbons are being explored [2,3,4,5,6,7]. In particular, one-dimensional semiconducting NFs produced by electrospinning have several advantages, such as ease of processing, large-area manufacturing, readily controllable doping, high surface-to-volume ratio, and high flexibility [8,9,10]. Accordingly, several research teams are actively developing methods for fabricating metal oxide semiconductors with excellent optical and electrical properties in the form of NFs with the aim of using them to produce the channels in field-effect transistors (FETs) [11,12]. However, the as-spun NFs collected on the collector electrode are unsuitable for use as platforms in various applications because of their poor uniformity, large diameter, high impurity concentration, and low surface-area-to-volume ratio [13]. In addition, NF-based electronic devices that contain organic solvents and charge trap sites can exhibit unsatisfactory performance. Therefore, a calcination process for eliminating the impurities present and reducing the diameter of the electrospun oxide semiconductor NFs is considered mandatory for improving the electrical and physical properties of the devices based on them [14,15]. Microwave annealing (MWA), which is a fast, cost-effective, and highly efficient method, is expected to find wide use as a next-generation thermal processing technology and is already being used in various fields [16,17]. Nevertheless, there have been few reports of the effects of microwave processing on the calcination of NFs. Meanwhile, plasma surface treatments have been employed for improving specific properties of metal oxide semiconductors. Thus, it is expected that they would also improve the electrical properties of NFs even at low temperatures by modifying their surface chemical bonds [18,19,20,21,22]. In particular, oxygen plasma treatments can improve device performance by removing the organic moieties and defects present in the channel layer when it is formed by a solution-based process, such as electrospinning [23].

## 2. Materials and Methods

### 2.1. Synthesis of IGZO/PVP Precursor Solution

The precursor for the indium gallium zinc oxide (IGZO) NFs was synthesized via a sol-gel reaction. The IGZO precursor solution was prepared by dissolving indium nitrate hydrate [In(NO_3_)_3_·*x*H_2_O, 99.99%], gallium nitrate hydrate [Ga(NO_3_)_3_·*x*H_2_O, 99.99%], and zinc acetate dehydrate [Zn(CH_3_COO)_2_·2H_2_O, 99.99%] in 0.5 mL *N*, *N*-dimethylformamide in a molar ratio of 2:1:1. The mixture was stirred in a closed vessel with a magnetic stirrer using an electronic agitator at 800 rpm for 2 h at 20 °C. Subsequently, 2.5 mL of ethanol containing 0.18 g of polyvinylpyrrolidone (PVP, M_W_ ≈ 1,300,000) was blended with the IGZO solution. This was followed by stirring in an agitator for 2 h at 20 °C. All the reagents were purchased from Sigma Aldrich (Saint Louis, MO, USA) and used without further purification.

### 2.2. Fabrication of IGZO NF FETs

The p-type bulk Si wafers used as the substrates were cleaned by standard wet chemical cleaning based on the RCA method. Different high-*k* dielectrics (Al_2_O_3_, HfO_2_, ZrO_2_, and Ta_2_O_5_) as well as SiO_2_ were deposited to a layer thickness of 100 nm for use as the bottom-gate dielectric layer. This was performed using radio-frequency magnetron sputtering (ULTECH, Daegu, Republic of Korea). To determine the dielectric constants of the various dielectrics, metal–insulator–metal (MIM) capacitors with an Al‒dielectric‒Ti structure were fabricated to perform capacitance–voltage (CV) measurements. The IGZO NFs were then electrospun onto the gate dielectric layer, which served as the channel region of the FETs. This was performed using the prepared blend of the IGZO precursor and the PVP-based solution.

A schematic diagram of the electrospinning of the IGZO NFs is shown in Figure 1a. The electrospinning process was as follows: The IGZO/PVP precursor solution was loaded into a plastic syringe, which was fixed to the syringe pump (NE-1000, New Era Pump Systems Inc., Farmingdale, NY, USA). The flow of the syringe pump was maintained at 0.4 mL/h, the pinhead internal diameter of the needle was 0.635 mm, and the vertical distance (work distance) from the needle to the grounded collector (Cu plate) was 20 cm. The positive voltage was adjusted to 20 kV, and the relative humidity and temperature were kept constant at 20% and 20 °C, respectively. The as-spun IGZO NFs that collected on the high-*k* gate dielectric layers were subjected to MWA calcination. In order to remove the solvent and impurities present in the IGZO/PVP composites and improve their electrical properties by reducing the number of trap sites, they were subjected to microwave irradiation at 2.45 GHz and 1800 W for 2 min. The ambient in the MW chamber was controlled using O_2_ gas, which is effective for removing organic solvents and impurities. In addition, for comparison, calcination was also performed using conventional furnace annealing (CFA) at 600 °C in an O_2_ gas ambient for 30 min.

Figure 2a shows a schematic of the microwave irradiation system used in this study, while Figure 2b shows the temperature profiles for the MWA and CFA processes used for calcining the electrospun IGZO NFs. Compared to the CFA method, the MWA method was shorter and had a higher ramp-up/down rate; this can significantly lower the thermal budget, which is expressed as the product of the process time and temperature. Since it is difficult to measure the temperature within the MW chamber using metal thermocouples, we monitored the temperature during the microwave irradiation process in real time using an infrared (IR) thermometer. The temperature of the calcined sample for a microwave output of 1800 W was found to be approximately 530 °C. Therefore, the thermal budget for the calcination process was determined as the areas of the temperature profiles shown in Figure 2b. It can be seen that the MWA process had a lower thermal budget at 7.3 × 10^4^ °C·s, while that of the CFA process was 2.23 × 10^6^ °C·s. The calcined samples were then exposed to an oxygen plasma at a pressure of 300 mTorr and power of 200 W for 20 s using a reactive ion etching system. The active region of the IGZO NF FETs, which had a channel width (W) and length (L) of 20 and 10 μm, respectively, was defined by photolithography and wet etching using a 30:1 buffer oxide etchant. Finally, a 150 nm-thick Ti film was deposited using an e-beam evaporator, and the source and drain electrodes were formed by the lift-off method to fabricate bottom-gate-type IGZO NF FETs, as shown in Figure 1b.

### 2.3. Characterization Methods

The surface morphology of the IGZO NFs was analyzed using scanning electron microscopy (SEM; Sirion 400, FEI Company, Hillsboro, OR, USA). The optical properties of the IGZO NFs were investigated using an Agilent 8453 ultraviolet-visible (UV-vis) spectrophotometer (Agilent technologies, Wilmington, DE, USA) in the wavelength range of 190–1100 nm. The chemical composition of the IGZO NFs was analyzed by X-ray photoelectron spectroscopy (XPS; PHI 5000 Versa Probe II, ULVAC, Chigasaki, Kanagawa, Japan), performed using monochromatic Al-Kα radiation (λ = 0.833 nm). The electrical performances of the IGZO NF-based FETs were measured using an Agilent 4156B precision semiconductor parameter analyzer (Agilent technologies, Wilmington, DE, USA). The measurements were performed in a dark box to block the external light and electrical noise.

## 3. Results and Discussion

### 3.1. Structural and Optical Properties of IGZO NFs

The morphological properties of the electrospun IGZO NFs were analyzed using SEM analysis. Figure 3a is an SEM image of the as-spun NFs while Figure 3b–e are SEM images of the CFA- or MWA-calcined NFs after the oxygen plasma treatment. It can be seen that the NFs have a uniform morphology and random network-like structure. It was found that the as-spun NFs were larger than 900 nm in diameter because they contained PVP and other organic solvent components. However, when the as-spun NFs were exposed to thermal energy, low-molecular-weight minerals such as pyrrolidone were released into the atmosphere, resulting in a decrease in the NF diameter. After the CFA calcination process at 600 °C (see Figure 3b) or the MWA calcination process at 1800 W (see Figure 3d), the diameter of the IGZO NFs decreased. In addition, it can be seen that the diameter of the MWA-calcined NFs was smaller than that of the CFA-calcined NFs. In order to confirm the diameter of the electrospun IGZO NF, we randomly selected 20 NFs from the SEM image and confirmed the diameter. Through their average, the average diameter of NF was calculated and shown in Figure 3f. Therefore, from the morphological properties of the electrospun IGZO NFs, it was confirmed that the MWA process has a superior calcination effect (7.3 × 10^4^ °C·s versus. 2.23 × 10^6^ °C·s) despite having a lower thermal budget than that of the CFA process. Moreover, the postcalcination oxygen plasma treatment further reduced the diameter by effectively removing the impurities from the electrospun IGZO NFs.

Figure 4a shows the transmittance spectra of the electrospun IGZO NFs subjected to the two calcination processes and the postcalcination oxygen plasma surface treatment, while Figure 4b shows their average transmittance in the visible light region (400–700 nm). The as-spun IGZO NFs show the lowest average transmittance at 73.6%. This was because they contained the polymer matrix and solvent in high contents. On the other hand, the MWA- and CFA-calcined NFs show significantly higher transmittances, owing to the removal of the solvent and impurities. In addition, the transmittance of the MWA-calcined NFs is higher than that of the CFA-calcined NFs. Finally, the postcalcination oxygen plasma treatment was more effective in improving the transmittance. This is the reason that the IGZO NFs subjected to MWA calcination followed by the oxygen plasma treatment exhibited the highest average transmittance at 93.4%.

### 3.2. Electrical Properties of IGZO NF FETs

The electrical properties of the bottom-gate-type IGZO NF FETs formed using the different high-*k* gate dielectrics and subjected to MWA or CFA calcination and a subsequent oxygen plasma treatment were evaluated. Figure 5a,b show the transfer characteristics (I_D_–V_G_), while Figure 5c,d show the output characteristics (I_D_–V_D_) of the IGZO NF FETs. It can be seen that the devices based on the MWA-calcined NFs show better electrical performance than those based on the CFA-calcined NFs. In addition, the postcalcination oxygen plasma treatment clearly improved the electrical properties of the IGZO NF FETs, regardless of the type of gate dielectric used. This is because together, the MWA calcination process and the oxygen plasma treatment could remove most of the impurities and charge traps from the electrospun IGZO NFs [24]. In particular, it can be seen that the device based on the NFs and Ta_2_O_5_ as the dielectric exhibited excellent performance. The dielectric constants of SiO_2_, Al_2_O_3_, HfO_2_, ZrO_2_, and Ta_2_O_5_ were 3.8, 8.9, 17.7, 22.3, and 26.9, respectively, as determined from the C–V curves of the fabricated MIM capacitors, as shown in Figure 6. Therefore, the drain current of the IGZO NF FETs increased when a dielectric with a higher constant was used for the gate insulator. The values of the other electrical parameters such as the threshold voltage (*V_TH_*), subthreshold swing (*SS*), and ON/OFF current ratio (*I_ON_*/*I_OFF_*) could be determined from the transfer characteristic curves and field effect mobility (*μ_FE_*) is defined by Equation (1) where *g_m_* is transconductance, *C_ox_* is oxide capacitance, and *V_DS_* is applied drain voltage [25]. There is also an alternative method commonly used to understand the trapping mechanism and estimate the electrical mobility of FETs composed of low mobility channel such as 2D thin-film geometry and organic FET by the space charge limited current (SCLC) [26,27,28]. Meanwhile, most of the reports related to oxide semiconductor-based FETs have evaluated the field-effect mobility using a drift-diffusion transport mechanism. These electrical parameters are listed in Table 1. With respect to the calcination method, it can be seen that the MWA-calcined FETs showed better electrical properties than those of the CFA-calcined devices, in that the former exhibited a smaller SS, higher *μ_FE_*, and larger *I_ON_*/*I_OFF_* value. With respect to the gate dielectric, the device based on SiO_2_ showed the poorest electrical properties in terms of the *SS*, *μ_FE_*, and *I_ON_*/*I_OFF_*. This was true regardless of the calcination method used. Meanwhile, with respect to the various high-*k* insulating materials evaluated in this study, the *SS*, *μ_FE_*, and *I_ON_*/*I_OFF_* values of the devices based on them could be arranged in the following order: Al_2_O_3_, HfO_2_, ZrO_2_, and Ta_2_O_5_. This result is because the higher the dielectric constant of the gate dielectric, the more the electric field is transferred from the gate to the channel of the FET. When the electric field is strongly transmitted to the channel, the concentration of electrons induced in the channel increases. Therefore, the higher the dielectric constant, the better the device performance.
(1)$$\mu_{FE}=g_{m}\frac{L}{WC_{ox}V_{DS}}$$

In addition, we calculated the interface trap density (*D_it_*) from the *SS* value determined from the transfer characteristic curves and the equation given below [29]. The results are shown in Figure 7.
(2)$$D_{it}=\frac{SS\;C_{i}\;log\left(e\right)}{q\;k_{B}\;T}$$
where *q* is the electron charge, *k_B_* is the Boltzmann constant, *T* is the absolute temperature, and *C_i_* is the capacitance of the gate dielectric per unit area. It can be seen that, regardless of the type of oxide film used, the MWA-calcined FETs (see Figure 5b) exhibited lower *D_it_* values than those of the CFA-calcined devices (see Figure 5a). In addition, the postcalcination oxygen plasma treatment was also effective in reducing *D_it_*. For instance, it reduced the *D_it_* value of the MWA-calcined devices even further. Therefore, it can be concluded that the MWA calcination process and postcalcination oxygen plasma treatment were effective in reducing the *D_it_* value of the IGZO NF FETs based on the different high-*k* gate dielectrics.

Next, we used the fabricated electrospun IGZO NF FETs to produce a resistor-loaded-inverter in order to further analyze the effects of the calcination process, postcalcination oxygen plasma treatment, and gate dielectric used. The static voltage transfer characteristics (VTC) of the resistor-loaded inverters were measured using a resistor of 370 MΩ, which was connected in series to the electrospun IGZO NF FET being tested. Figure 8a,b show the VTC (V_OUT_–V_IN_) curves (V_DD_ = 10 V), while Figure 8c,d show their gains as determined from the VTC curves. Gain refers to the amount of change in V_OUT_ with respect to V_IN_ in the VTC and is defined as $\partial V_{\mathrm{OUT}}/\partial V_{\mathrm{IN}}$. For both types of devices, the V_OUT_ value for a low V_IN_ was almost equal to the applied V_DD_. On the other hand, when V_IN_ was increased, the MWA-calcined devices exhibited suitable inverting characteristics, while the CFA-calcined devices did not. In addition, as shown in Figure 8d, the gain was higher for the cases where the plasma treatment was not performed, meaning that the operation of the inverter was improved by the plasma treatment. In particular, the greater the dielectric constant of the gate dielectric used (as in the cases of Ta_2_O_5_ and ZrO_2_), the higher the gain of the inverter. This, in turn, was closely related to the electrical properties shown in Figure 5 and Table 1.

### 3.3. Stability of IGZO NF FETs

The instability of the long-term operation of metal oxide semiconductor-based electronic devices is a significant problem. Although this problem is more prominent in NF-type metal oxide semiconductors because of the presence of a large number of defects [30,31], there is a lack of studies on the stability of metal oxide semiconductor-based NF FETs. In addition, in general, even when an annealing process is performed, a high-*k* gate dielectric will contain more defects than SiO_2_. Furthermore, the characteristics of the interface between the gate dielectric and the channel layer will be poor [32]. Therefore, we investigated the effects of MWA calcination and the oxygen plasma treatment on the long-term electrical stability of the IGZO NF FETs based on the different high-*k* gate dielectrics. This was performed by measuring the changes in the threshold voltage (ΔV_TH_) under a positive bias temperature stress (PBTS) and a negative bias temperature stress (NBTS).

Figure 9 shows the time dependence of ΔV_TH_ with respect to PBTS and NBTS for the devices subjected to the calcination process and oxygen plasma treatment. These PBTS and NBTS tests were performed at *V_G_* = *V_TH_*_0_ ± 5 V and *V_D_* = 0 V for 10^3^ s at 25, 55, and 85 °C; here, *V_TH_*_0_ is the initial V_TH_ before the application of the gate bias stress. We found that *V_TH_* shifted to the right in the case of a PBTS and to the left for a NBTS, as the stress time was increased. In addition, Δ*V_TH_* became larger with increasing temperature. Δ*V_TH_* was positive during the PBTS test because the adsorbed oxygen molecules (acceptor like) formed a depletion layer below the active channel region. On the other hand, Δ*V_TH_* was negative during the NBTS test because the oxygen vacancies (donor like) formed an accumulation layer below the active channel region [30,31,33,34]. It can be seen that the Δ*V_TH_* value of the MWA-calcined devices was smaller than that of the CFA-calcined devices. In particular, it is worth noting that the postcalcination oxygen plasma treatment decreased the Δ*V_TH_* value of the electrospun IGZO NF FETs. With respect to the gate dielectrics, the device based on SiO_2_ exhibited the lowest Δ*V_TH_* and the most stable operation. In the case of the high-*k* dielectrics, the device based on Ta_2_O_5_ was the most stable, while that based on HfO_2_ was the most unstable.

Figure 9 shows the fitting results for the experimental data showing the time dependence of Δ*V_TH_* during the PBTS and NBTS tests. As stated above, the data were fitted using the stretched-exponential equation shown below, which showed that the changes in Δ*V_TH_* were caused by the thermally activated charge trapping mechanism [35].
(3)$$\Delta V_{TH}\left(t\right)=\Delta V_{TH0}\left[1-exp\left\{-{\left(\frac{t}{\tau }\right)}^{\beta }\right\}\right]$$
where Δ*V_TH_*_0_ is the threshold voltage shift (i.e., Δ*I*) at the initial time, *τ* is the characteristic trapping time of the charge carrier, and *β* is the stretched-exponential exponent. Based on the stretched-exponential equation, the *τ* value of a thermally activated carrier can be expressed as follows:(4)$$\tau =\tau_{0}exp\left(\frac{E_{\tau }}{k_{B}T}\right)=\nu^{-1}exp\left(\frac{E_{\tau }}{k_{B}T}\right)$$
where the thermal activation energy is given by *E_a_* = *E_τ_β*. Furthermore, *E_τ_* is the average effective energy barrier that must be overcome before the electrons in the IGZO NF channel enter the gate dielectric, *τ*_0_ is the thermal prefactor, and ν is the frequency prefactor for emission across the barrier.

The fitting curves in Figure 9 agree well with the measured data. The values of τ calculated using the equation are plotted in Figure 10. As the temperature during the PBTS and NBTS tests was increased, the charge trapping time, *τ*, decreased gradually. This means that *τ* is temperature dependent. In addition, the MWA-calcined IGZO NF FETs exhibited higher *τ* values than those of the CFA-calcined devices. This means that the MWA-calcined IGZO NF FETs captured carriers more slowly than the CFA-calcined devices, resulting in better long-term stability. In addition, the postcalcination oxygen plasma treatment further improved τ and, in turn, the long-term stability of the electrospun IGZO NF FETs. Finally, with respect to the high-*k* dielectrics (excluding SiO_2_), the device based on Ta_2_O_5_ showed the highest *τ* value, indicating that this device had the most stable operation.

Figure 11 shows the relationship between ln(*τ*) and the inverse of the temperature (1/*T*) for the IGZO NF FETs during the PBTS and NBTS tests. The fact that the relationship between ln(*τ*) and 1/*T* was linear suggests that charge trapping in the IGZO NF FETs is associated with thermal activation. The slope of the Arrhenius plot represents the average effective height of the barrier (*E_τ_*) for charge transport, which is related to the lattice arrangement within the channel. A lower *E_τ_* means a channel with a lower defect density and a better lattice arrangement and hence a more stable device [36]. It is interesting to note that the I value of the MWA-calcined IGZO NF FETs is lower than that of the CFA-calcined devices. In addition, it was found that the postcalcination oxygen plasma treatment further reduced *E_τ_*. Therefore, these results further confirm that MWA calcination is more effective in improving the stability of electrospun IGZO NF FETs than CFA calcination and that the subsequent oxygen plasma treatment further enhances device stability.

### 3.4. Chemical Properties of Electrospun IGZO NFs

Figure 12 shows the XPS O1 spectra of the IGZO NFs after (a) CFA calcination and (b) MWA calcination. XPS was performed to determine the chemical compositions of the NFs, and the spectra were measured after etching the surfaces of the IGZO NFs with Ar^+^ ions to a depth of a few nanometers. This was performed to reduce the error owing to surface contaminants. Taking into account the precursor solution present in the electrospun IGZO NFs, we deconvoluted the O1s peak to those corresponding to the individual chemical components. The main peak at 529 eV represents stoichiometric oxygen (M-O), the subpeak at 530 eV represents the oxygen vacancies (M-O_vac_), and the subpeak at 531 eV is associated with loosely bound oxygen impurities (M-OH), such as chemically adsorbed oxygen, H_2_O, and CO_3_ [37].

Table 2 lists the chemical compositions of the calcined NFs as determined before and after the oxygen plasma treatment. It can be seen that the proportions of M-O_vac_ and M-OH were higher and that of M-O much smaller in the case of the MWA-calcined NFs as compared with those for the CFA-calcined NFs. In addition, after the plasma treatment, the proportion of M-O increased, while those of M-O_vac_ and M-OH decreased further. It is known that the oxygen deficiency associated with M-O_vac_ binding decreases the stability during the PBTS and NBTS tests, while loosely coupled oxygen impurities associated with M-OH binding act as charge traps and reduce the on-current of FETs [33,34]. Hence, we believe that the MWA calcination process and the subsequent oxygen plasma treatment are highly effective for improving the electrical properties and stability of electrospun IGZO NF FETs. The MWA process calcinates the NFs at a low thermal budget, while the oxygen plasma treatment effectively removes the impurities from the NFs.

## 4. Conclusions

In this study, we investigated the effects of different high-*k* gate dielectrics, MWA calcination, and a postcalcination oxygen plasma treatment on the properties of electrospun IGZO NF FETs. We used a number of high-*k* gate dielectrics such as Al_2_O_3_, HfO_2_, ZrO_2_, and Ta_2_O_5_ to ensure high-performance operation and fabricated n-type FETs by electrospinning the IGZO NF-based channel. The MWA process and the oxygen plasma treatment were performed sequentially to improve the electrical properties and stability of the electrospun IGZO NF FETs. We found that the higher the dielectric constant of the gate dielectric used, the better the performance of the IGZO NF FETs, with the device based on Ta_2_O_5_ exhibiting the best performance. In addition, the MWA-calcined FETs were superior to the CFA-calcined ones in terms of their electrical characteristics. Furthermore, the voltage transfer characteristics of the resistor-loaded inverter based on the former were also better. The postcalcination oxygen plasma treatment further improved the performance of these devices. The results of the PBTS and NBTS tests confirmed that the MWA calcination process and postcalcination oxygen plasma treatment significantly improved the stability of the IGZO NF FETs by reducing the numbers of defects and charge traps. In addition, it was found that the morphological and optical properties of the electrospun IGZO NFs were also improved by the MWA calcination process and postcalcination oxygen plasma treatment. A chemical analysis of the NFs using XPS showed that the calcination process and plasma treatment effectively removed the organic solvents and impurities that act as charge traps and were thus effective in improving the morphological, optical, and chemical properties of electrospun IGZO NFs. Hence, these processes should aid the development of high-performance IGZO NF FETs based on high-*k* gate dielectrics for use in next-generation electronic devices.

## Figures and Tables

**Figure 1 nanomaterials-10-01804-f001:**
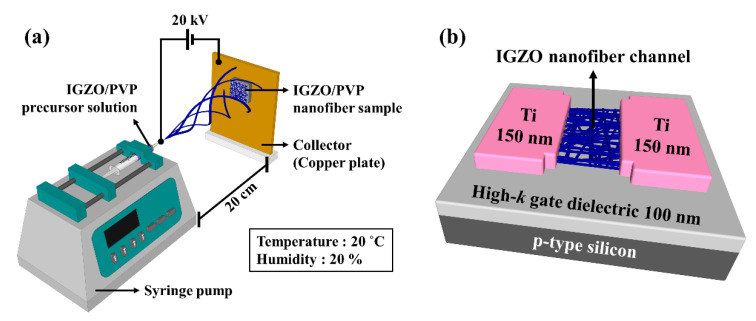
Schematic diagram of (**a**) electrospinning apparatus and (**b**) bottom-gate-type indium gallium zinc oxide (IGZO) nanofiber (NF) field-effect transistors (FETs) with high-*k* gate dielectrics.

**Figure 2 nanomaterials-10-01804-f002:**
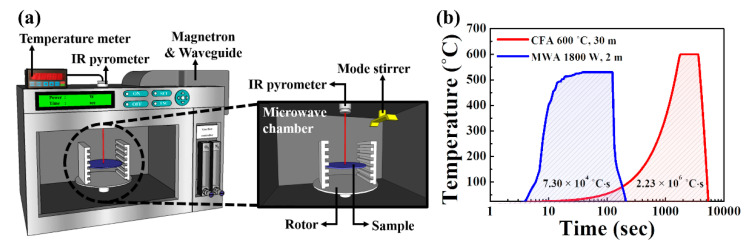
(**a**) Schematic of microwave irradiation system. (**b**) Temperature profiles of microwave annealing (MWA) and conventional furnace annealing (CFA) calcination processes.

**Figure 3 nanomaterials-10-01804-f003:**
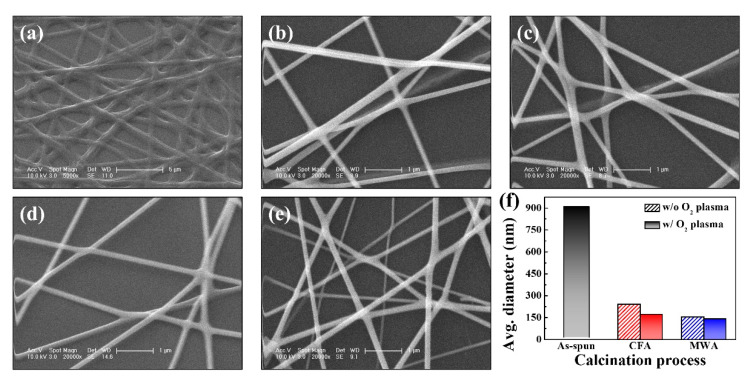
SEM images of electrospun IGZO NFs: (**a**) as-spun (×5000); CFA-calcined NFs (**b**) before and (**c**) after oxygen plasma treatment (×20,000); MWA-calcined NFs (**d**) before and (**e**) after oxygen plasma treatment (×20,000). (**f**) Average diameters of 20 fibers randomly selected from SEM images.

**Figure 4 nanomaterials-10-01804-f004:**
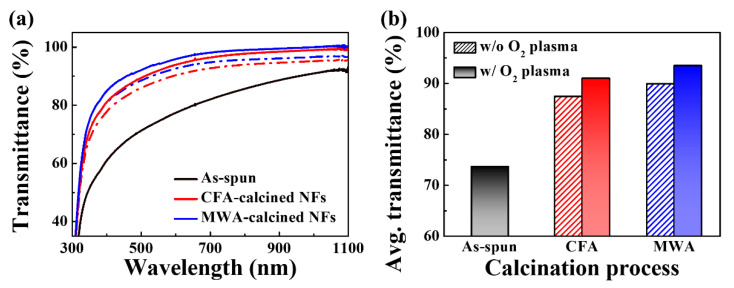
(**a**) Transmittance spectra and (**b**) average transmittances in visible light of electrospun IGZO NFs formed on glass substrates before and after calcination and oxygen plasma treatment.

**Figure 5 nanomaterials-10-01804-f005:**
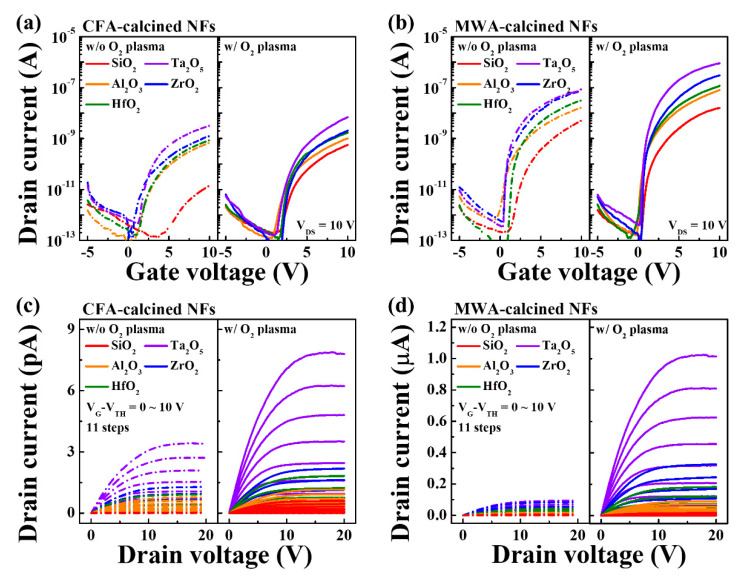
Electrical properties of bottom-gate-type IGZO NF FETs with different high-*k* dielectrics. (**a**,**b**) are curves of transfer characteristics (I_D_–V_G_) and (**c**,**d**) are curves of output characteristics (I_D_–V_D_). Dashed and solid lines indicate properties of FETs before and after postcalcination oxygen plasma treatment, respectively.

**Figure 6 nanomaterials-10-01804-f006:**
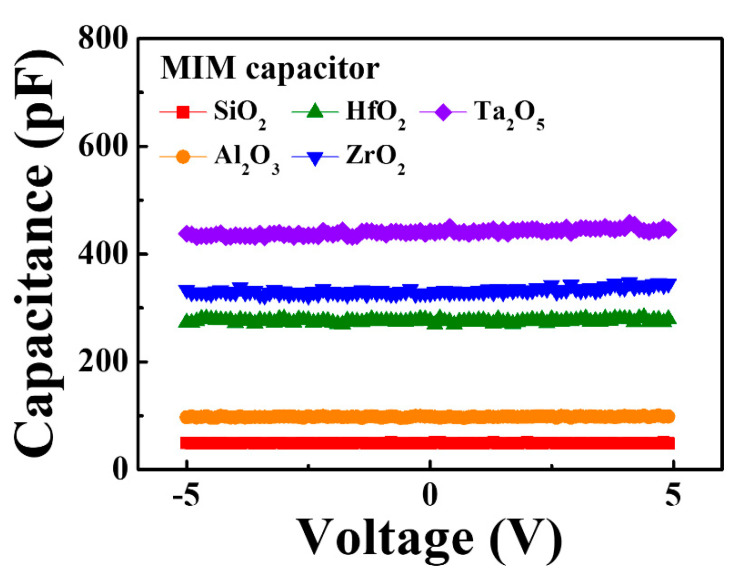
C-V characteristic curves of metal–insulator–metal (MIM) capacitor with various gate dielectrics.

**Figure 7 nanomaterials-10-01804-f007:**
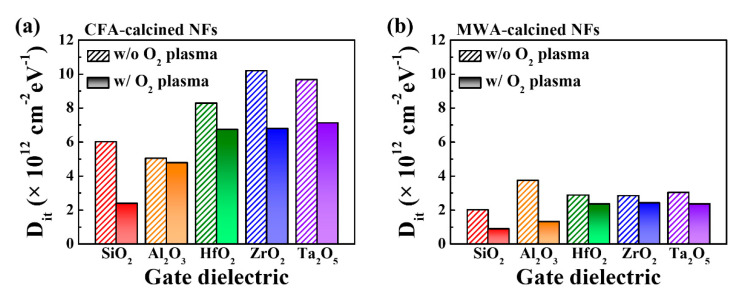
Interface trap density (*D_it_*) values of (**a**) CFA- and (**b**) MWA-calcined IGZO NF FETs. Dashed and solid bars indicate values obtained before and after postcalcination oxygen plasma treatment, respectively.

**Figure 8 nanomaterials-10-01804-f008:**
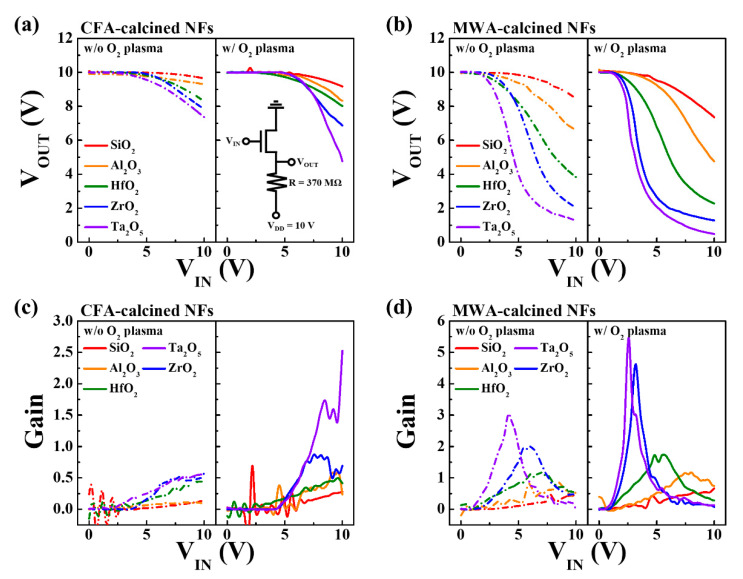
Voltage transfer characteristic (VTC) curves of resistor-loaded inverters based on (**a**) CFA- and (**b**) MWA-calcined IGZO NF FETs with different high-*k* dielectrics. Gains of inverters with (**c**) CFA- and (**d**) MWA-calcined IGZO NF FETs. Dashed and solid lines represent data for FETs obtained before and postcalcination oxygen plasma treatment, respectively.

**Figure 9 nanomaterials-10-01804-f009:**
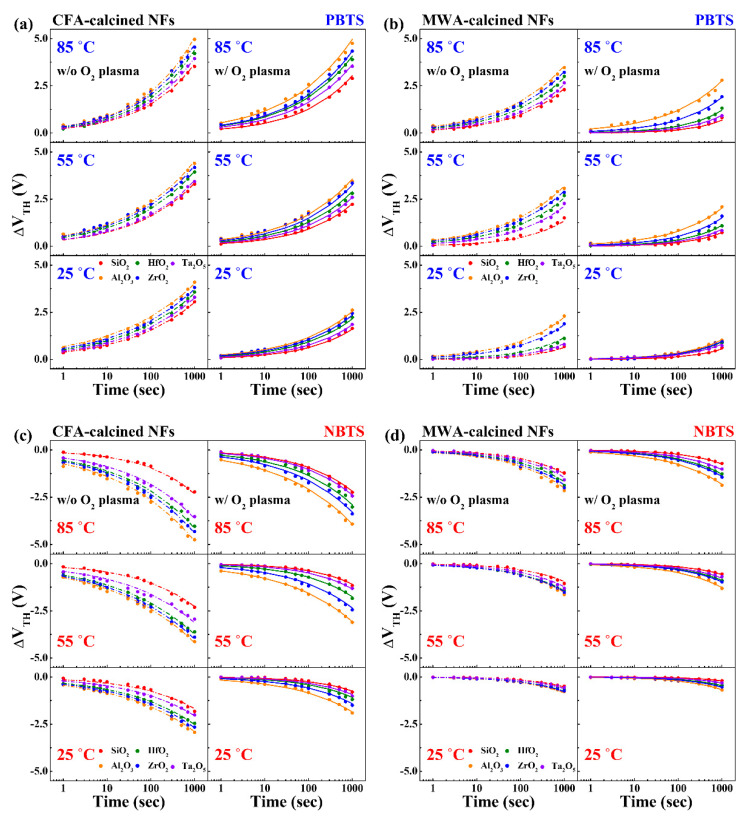
ΔV_TH_ values of IGZO NF FETs as functions of stress time for different high-*k* dielectrics. (**a**) CFA- and (**b**) MWA-calcined devices during PBTS test and (**c**) CFA- and (**d**) MWA-calcined devices during NBTS test. Dashed and solid lines are fitting curves for data obtained before and after postcalcination oxygen plasma treatment, respectively. Curves were fitted using stretched-exponential equation.

**Figure 10 nanomaterials-10-01804-f010:**
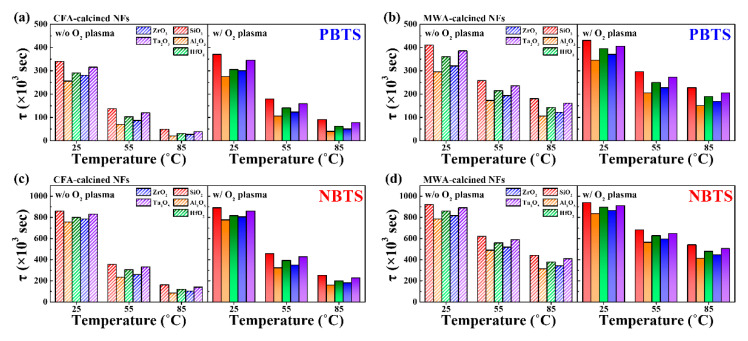
Charge trapping time (*τ*) values of IGZO NF FETs based on different high-*k* dielectrics as determined from PBTS and NBTS tests. (**a**) CFA-calcined and (**b**) MWA-calcined devices during PBTS test. (**c**) CFA-calcined and (**d**) MWA-calcined devices during NBTS test. Dashed and solid bars indicate data for FETs obtained before and after postcalcination oxygen plasma treatment, respectively.

**Figure 11 nanomaterials-10-01804-f011:**
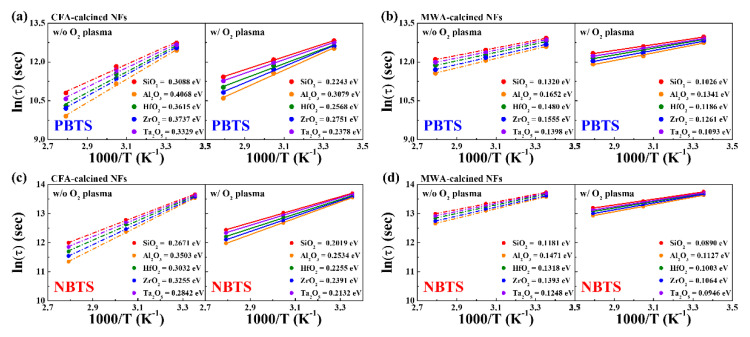
Logarithmic plots of charge trapping time (*τ*) against inverse of temperature (1/*T*) for IGZO NF FETs based on different high-*k* dielectrics. (**a**) CFA-calcined and (**b**) MWA-calcined devices during PBTS test. (**c**) CFA-calcined and (**b**) MWA-calcined devices during NBTS test. Dashed and solid lines are fitting curves for data obtained before and after postcalcination oxygen plasma treatment, respectively.

**Figure 12 nanomaterials-10-01804-f012:**
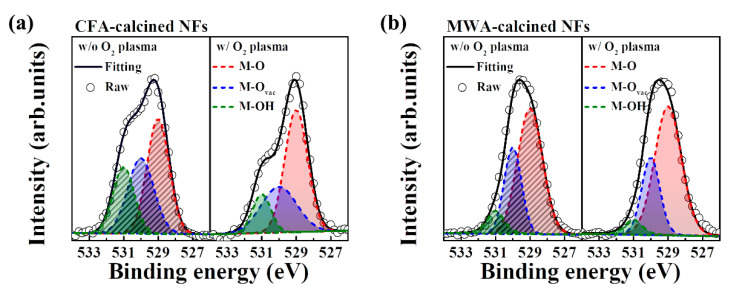
XPS O1s spectra of (**a**) CFA-calcined and (**b**) MWA-calcined IGZO NFs. Dashed and solid areas indicate data for IGZO NFs obtained before and after postcalcination oxygen plasma treatment, respectively.

**Table 1 nanomaterials-10-01804-t001:** Electrical parameters of IGZO NF FETs based on different high-*k* gate dielectrics.

Method for Calcinating NFs	Gate Dielectric	w/o O_2_ Plasma Treatment				w/ O_2_ Plasma Treatment			
		*V_TH_* (V)	*SS* (mV/dec)	*μ_FE_* (cm^2^/V^·^s)	*I_ON_*/*I_OFF_*	*V_TH_* (V)	*SS* (mV/dec)	*μ_FE_* (cm^2^/V^·^s)	*I_ON_*/*I_OFF_*
CTA	SiO_2_	8.6	1703.8	1.27 × 10^−4^	0.07 × 10^3^	3.1	676.1	1.69 × 10^−3^	2.82 × 10^3^
	Al_2_O_3_	2.6	610.7	8.25 × 10^−4^	3.42 × 10^3^	2.2	576.1	2.49 × 10^−3^	5.12 × 10^3^
	HfO_2_	2.4	504.6	1.28 × 10^−3^	4.31 × 10^3^	2.4	409.4	4.02 × 10^−3^	8.42 × 10^3^
	ZrO_2_	1.6	491.3	2.07 × 10^−3^	5.10 × 10^3^	2.7	327.8	4.47 × 10^−3^	1.02 × 10^4^
	Ta_2_O_5_	1.9	387.4	2.21 × 10^−3^	1.59 × 10^4^	2.1	284.9	6.38 × 10^−3^	3.46 × 10^4^
MWA	SiO_2_	2.2	568.4	2.48 × 10^−2^	0.25 × 10^4^	0.9	250.6	9.92 × 10^−2^	7.93 × 10^4^
	Al_2_O_3_	0.4	453.1	4.26 × 10^−2^	8.20 × 10^4^	0.2	160.1	1.34 × 10^−1^	4.01 × 10^5^
	HfO_2_	1.3	162.9	6.54 × 10^−2^	1.58 × 10^5^	0.3	155.7	2.06 × 10^−1^	5.82 × 10^5^
	ZrO_2_	0.6	127.3	1.09 × 10^−1^	3.83 × 10^5^	0.6	125.7	4.96 × 10^−1^	1.51 × 10^6^
	Ta_2_O_5_	0.5	113.8	1.31 × 10^−1^	4.37 × 10^5^	0.4	102.6	6.55 × 10^−1^	4.51 × 10^6^

**Table 2 nanomaterials-10-01804-t002:** Chemical compositions of electrospun CFA- and MWA-calcined IGZO NFs as determined before and after postcalcination oxygen plasma treatment.

	w/o O_2_ Plasma Treatment			w/ O_2_ Plasma Treatment		
	M-O (%)	M-O_vac_ (%)	M-OH (%)	M-O (%)	M-O_vac_ (%)	M-OH (%)
CFA calcined	42.8	33.3	23.9	54.5	31.7	13.8
MWA calcined	61.6	29.1	9.3	66.4	27.4	6.2

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
