# Peer review of "Performance Enhancement of Electrospun IGZO-Nanofiber-Based Field-Effect Transistors with High-k Gate Dielectrics through Microwave Annealing and Postcalcination Oxygen Plasma Treatment"

_nanomaterials, 2020, doi:10.3390/nano10091804_

Round 1

Reviewer 1 Report

I suggest to the authors to improve the quality of Figure 8.

Author Response

[Comment 1]

I suggest to the authors to improve the quality of Figure 8.

[Answer 1]

Thank you very much for your kind consideration. We resized the figure so that its contents can be seen clearly.

Reviewer 2 Report

See uploaded file

Author Response

[Comment 1]

The abstract of the manuscript is rather lengthy and describes, in details, the experimental/characterization procedures. However, the abstract does not seem to have sufficiently conveyed the key findings and the novelties of this work. I recommend the Authors to revise the abstract and to make it more succinct. Particularly, a stronger emphasis should be made to more prominently highlight the key findings of this manuscript so to justify the potential importance of this work in IGZO-based FET, rather than providing a lengthy description of the experimental procedures.

[Answer 1]

We appreciate your helpful comments. We revised the abstract to reveal key findings and novelties rather than explaining the experimental procedure of this work.

In response to the reviewer’s comments, we have also revised the manuscript in the following section for the reader's understanding:

(Line 11 – 26, Page 1 in revised manuscript)

We investigated the effects of various high-k gate dielectrics as well as microwave annealing (MWA) calcination and a post-calcination oxygen plasma treatment on the electrical properties and stability of electrospun indium gallium zinc oxide (IGZO)-nanofiber (NF)-based field-effect transistors (FETs). We found that the higher the dielectric constant of the gate dielectric, the better the electric field is transferred, resulting in better performance of the IGZO NF FET. In addition, the MWA-calcined IGZO NF FET was superior to the conventional furnace annealing-calcined device in terms of the electrical properties of the device and the operation of resistor-loaded inverter, and it was proved that the oxygen plasma treatment further improved the performance. The results of the gate bias temperature stress test confirmed that the MWA calcination process and post-calcination oxygen plasma treatment greatly improved the stability of the IGZO NF FET by reducing the number of defects and charge traps. This is verified that the MWA calcination process and oxygen plasma treatment effectively remove the organic solvent and impurities that act as charge traps in the chemical analysis of NF using X-ray photoelectron spectroscopy. Furthermore, it was demonstrated through scanning electron microscopy and ultraviolet-visible spectrophotometer that the MWA calcination process and post-calcination oxygen plasma treatment also improve the morphological and optical properties of IGZO NF.

[Comment 2]

What is the average thickness of the IGZO NF in the FET devices?

[Answer 2]

In the case of nanofibers, it is considered more appropriate to indicate diameter rather than thickness. We randomly selected 20 fibers from the SEM image and checked their diameter. Through their average, the average diameter of the fiber was calculated as shown in Figure 3(f).

In response to the reviewer’s comments, we have also revised the manuscript in the following section for the reader's understanding:

(Line 15 – 18, Page 4 in revised manuscript)

In order to confirm the diameter of the electrospun IGZO NF, we randomly selected 20 NFs from the SEM image and confirmed the diameter. Through their average, the average diameter of NF was calculated and shown in Figure 3(f).

[Comment 3]

How are the electrical mobility of the FETs measured? A brief discussion, accompanied by key expressions used for mobility extraction, should be included in the manuscript.

[Answer 3]

Thank you for your valuable comment. We added an equation for field effect mobility to help readers understand following your comments.

In response to the reviewer’s comments, we have also revised the manuscript in the following section for the reader's understanding:

(Line 20 – 23, Page 5 in revised manuscript)

The values of the other electrical parameters such as the threshold voltage (VTH), subthreshold swing (SS), and ON/OFF current ratio (ION/IOFF) could be determined from the transfer characteristic curves and field effect mobility (μFE) is defined by equation (1) where gm is transconductance, Cox is oxide capacitance, and VDS is applied drain voltage [25].

[Comment 4]

From the electrical measurements in Fig. 5 and Table I, it is immediately clear that the gate dielectric has a strong influence on the electrical performance of the FET – gate dielectric with higher dielectric constant is expected to lead to better electrical mobility, SS and ON/OFF ratio. How does these quantities scale with the dielectric constants of different gate dielectric? The Authors should include a “companion” figure to Table I, plotting mobility, SS and ON/OFF ratio versus the dielectric constant, and briefly discuss whether any linear/nonlinear scaling behaviors are observed. Such figure might be useful for the dielectric engineering/design and optimization of IGZO-based FET in future works.

[Answer 4]

Thank you for your helpful comments. We plotted electrical parameters such as subthreshold swing (SS), field effect mobility (μFE) and ON/OFF current ratio (ION/IOFF) as a function of dielectric constant, which is shown in Figure R1. The red, orange, green, blue and purple symbols represent SiO2, Al2O3, HfO2, ZrO2, Ta2O5, respectively, and their dielectric constants are 3.9, 8.9, 17.7, 22.3, and 26.9, respectively. The open and closed circles represent the electrical parameters without and with oxygen plasma treatment, respectively. It is found that electrical parameters tend to improve with increasing dielectric constant. However, in some cases, the linearity (R2) of the data is not good as indicated by the fitting lines. Therefore, the assertion that “the improvement of dielectric constant and electrical properties is in a linear relationship” is less accurate and can lead to reader misunderstanding. For that reason, we believe that summarizing the electrical parameters in a table is appropriate to avoid further confusion. It would be appreciated if the reviewer would consider this aspect.

[Comment 5]

The interface trap density is analysed in Pg. 7 and it is demonstrated that O2 plasma treatment and MWA calcination are effective in reducing the interface trap density. Apart from using Eq. (1) which is based on the extracted SS and gate dielectric capacitance, the space-charge-limited current (SCLC) is also very commonly used to understand the trapping mechanisms and further offer an alternative methods to estimate the electrical mobility of FET composed of low-mobility channel [e.g. Phys. Rev. 103, 1648 (1956), Chem. Rev. 107, 926 (2007) and more recently for ultrathin-film FET in Phys. Rev. B 95, 165409 (2017)]. I recommend the Authors to briefly discuss SCLC as an alternative measurement technique for the electrical mobility and the charge trapping mechanism in low-mobility material.

[Answer 5]

The SCLC mechanism commented by the reviewer is also applied to extract the charge mobility of low-mobility materials such as 2D thin film geometries or organic FETs (OFETs). Meanwhile, most of the reports related to oxide semiconductor-based FETs have evaluated the field-effect mobility using a drift-diffusion transport mechanism. A number of our papers reported in the past have been evaluated in the same way. Therefore, it would be more desirable to evaluate the device performance in terms of field-effect mobility for a consistent comparison with other reports. In addition, the main purpose of this study is to compare the performance of IGZO NF FETs according to various gate dielectric materials. As reviewers commented, although SCLC is also very commonly used to understand the trapping mechanism and provide an alternative method of estimating the electrical mobility of FETs composed of low-mobility channel, we consider that the evaluation of field effect mobility will be a more consistent comparison with previous reports. Instead, referring to the reviewer's comment, we described in the manuscript that the SCLC is also commonly used to understand the trapping mechanism and provide an alternative method to estimate the electrical mobility of FET composed of low-mobility channel. It would be appreciated if you understand why we have evaluated and compared field-effect mobility in this manuscript.

In response to the reviewer’s comments, we have also revised the manuscript in the following section for the reader's understanding:

(Line 23 – 28, Page 5 in revised manuscript)

There is also alternative method commonly used to understand the trapping mechanism and estimate the electrical mobility of FETs composed of low mobility channel such as 2D thin-film geometry and organic FET by the space charge limited current (SCLC) [26,27,28]. Meanwhile, most of the reports related to oxide semiconductor-based FETs have evaluated the field-effect mobility using a drift-diffusion transport mechanism.

[Comment 6]

How are the gains in Fig. 7(c) and (d) calculated? It is mentioned that gain is mathematically defined as ?????/????. However, the actual method/expression used to estimate the gain should be quoted and briefly discussed in the manuscript. Also, how does the VTC and gain values of the IGZO-based inverter compare to other competing approaches in the literature? The Authors should provide a comparison with reported values in the literature so to better establish the potential novelty of their devices.

[Answer 6]

Gain refers to the amount of change in VOUT with respect to VIN in the voltage transfer curve (VTC), and is defined as  as you mentioned. Thanks to the reviewer's comments, we were able to revise the manuscript in more detail. On the other hand, VTC in resistor-loaded inverter depends on VDD and load resistance. The gain extracted from VTC also depends on VDD and load resistance. Therefore, it is difficult to accurately compare VTC and gain values with other literature because VDD and load resistance are different. We would appreciate if you focus on the "0" and "1" logic operation of IGZO NF FET based resistor-loaded inverter, rather than comparing exact values with other literature. We implemented a resister-loaded inverter using MWA-calcined IGZO NF FET with Ta2O5 gate dielectric and load resistance, and proved excellent inverting operation.

In response to the reviewer’s comments, we have also revised the manuscript in the following section for the reader's understanding:

(Line 1 – 2, Page 8 in revised manuscript)

Gain refers to the amount of change in VOUT with respect to VIN, and is defined as .

[Comment 7]

Can the Authors briefly comment on the issue of contact resistance in their devices? It is known that contact resistance is highly problematic in nanomaterial-based FET, especially when the channel is shortened towards nanoscale regime. Do the Authors expect the contact resistance to play an important role in the charge transport behavior of their device? If so, what can potentially be done to mitigate such effects?

[Answer 7]

Thank you very much for your important advice on our manuscript. The performance of FETs is affected by geometric factors, such as the channel length, active layer thickness, source/drain (S/D) series resistance, and metal contact characteristics associated with parasitic resistance effects. If these parasitic resistances are large, transistor performance deteriorates. In our previous work [S. K. Cho, W. J. Cho, Sci. Rep., 10(1), 1-9 (2020)], the contact resistance of S/D electrodes in IGZO NF FETs has already been evaluated using the transmission line method (TLM). It was demonstrated that oxygen plasma treatment improves the electrical properties and instability of IGZO NF FET as well as contact resistance. We would be grateful if reviewers refer to our previous research.

Reviewer 3 Report

This paper comprehensively investigated IGZO NF FETs using varies high-k dielectrics. I think the results are interesting and useful for the diverse TFT applications.

The manuscript is well written and after some minor revision could be suitable for publication.

1. Please also provide CV and IV curves of MIM capacitors with varies dielectrics.

2. Please add more description to explain why the TFTs using wider bandgap dielectrics such as SiO2 and Al2O3 showed poor on/off current ratios after oxygen plasma treatment.

Author Response

[Comment 1]

Please also provide CV and IV curves of MIM capacitors with varies dielectrics.

[Answer 1]

We appreciate your helpful comments. We revised the manuscript by adding a C-V curves that extracted the dielectric constant.

In response to the reviewer’s comments, we have also revised the manuscript in the following section for the reader's understanding:

(Line 16 – 18, Page 5 in revised manuscript)

The dielectric constants of SiO2, Al2O3, HfO2, ZrO2, and Ta2O5 were 3.8, 8.9, 17.7, 22.3, and 26.9, respectively, as determined from the C–V curves of the fabricated MIM capacitors as shown in Figure 5.

[Comment 2]

Please add more description to explain why the TFTs using wider bandgap dielectrics such as SiO2 and Al2O3 showed poor on/off current ratios after oxygen plasma treatment.

[Answer 2]

We appreciate your helpful comments. The higher the dielectric constant of the gate dielectric, the more the electric field is transferred from the gate to the channel of the FET. When the electric field is transmitted strongly into the channel, the concentration of electrons induced in the channel increases. Therefore, the higher the dielectric constant, the better the device performance. Unlike HfO2, ZrO2, and Ta2O5, SiO2 and Al2O3 have low dielectric constants of less than 10, so FETs using them show poor device performance.

In response to the reviewer’s comments, we have also revised the manuscript in the following section for the reader's understanding:

(Line 35 – 38, Page 5 in revised manuscript)

This result is because the higher the dielectric constant of the gate dielectric, the more the electric field is transferred from the gate to the channel of the FET. When the electric field is strongly transmitted to the channel, the concentration of electrons induced in the channel increases. Therefore, the higher the dielectric constant, the better the device performance.
